# Effect Modification of Race on the Associated Tumor Size at Diagnosis and 10-Year Cancer Survival Rates in Women with Cervical Squamous Cell Carcinoma in the United States

**DOI:** 10.3390/ijerph20186742

**Published:** 2023-09-12

**Authors:** Samara Khan, Tooba Sheikh, Grettel Castro, Noël C. Barengo

**Affiliations:** 1Division of Medical and Population Health Sciences Education and Research, Department of Medical Education, Herbert Wertheim College of Medicine, Florida International University, Miami, FL 33199, USA; tshei002@med.fiu.edu (T.S.); gcastro@fiu.edu (G.C.); nbarengo@fiu.edu (N.C.B.); 2Escuela Superior de Medicina, Universidad Nacional de Mar del Plata, Mar del Plata 7600, Argentina

**Keywords:** cervical squamous cell cancer, cervical cancer, squamous cell cancer, survival, tumor size, race, SEER, database

## Abstract

Background: While there may be an association between race, tumor size, and survival in patients with cervical squamous cell carcinoma (SCC), evidence on the effect of race on the association between tumor size at diagnosis and survival is limited. Our study evaluated whether race modifies the association between tumor size and 10-year survival in cervical SCC. Methods: This non-concurrent cohort study with the Surveillance, Epidemiology, and End Results (SEER) database included women diagnosed with cervical SCC between 2004–2018. The independent variable was diagnosis tumor size, where 2–4 cm was classified as high risk, and <2 cm was considered low risk. The dependent variable was 10-year cancer-specific survival rates, and race was our effect modifier. Unadjusted and adjusted Cox regression analysis were conducted to calculate hazard ratios (HR) and 95% confidence intervals (CI). Results: While a higher proportion of Black/Asian/Pacific Islander patients presented with tumor sizes of 2–4 cm compared to <2 cm (32.8% vs. 22.3%; *p* = 0.007) and having a tumor size of 2–4 cm had a significantly decreased 10-year survival (HR: 2.7; 95% CI: 1.3–5.8), the interaction between race and 10-year cancer-specific survival was not significant. Conclusion: Although race did not modify the interaction between tumor size and 10-year survival, emphasis needs to be placed on screening and proper data collection, especially for minority races, and studies with larger sample sizes should be conducted in order to better implement future recommendations to improve health and survival.

## 1. Introduction

The cervix is an organ that is located at the lower end of the uterus that forms a narrow connection between the uterus and the vagina. This connection is lined by stratified squamous epithelium because of the physical stress it must endure. This stratified squamous epithelium covers the exocervix and the mucous secreting columnar epithelium that is common to the endocervical canal. This transition is called the squamocolumnar junction, and it is believed to be the site with the highest risk of viral neoplastic transformation. The tumors that arise from the ectocervix are most commonly squamous cell carcinomas, and the most common cause of this is infection with the Human Papilloma Virus. Without treatment, persistent infection with Human Papilloma Virus (HPV) results in the premalignant conditions such as intraepithelial neoplasia and may progress to cervical squamous cell carcinoma (SCC) [1,2]. The prevalence of this disease can be associated with many different risk factors that may be more significant according to race. Socioeconomic factors such as access to healthcare and insurance, and other social determinants of health play an important role in the development of this condition. According to Wang et al., despite screening associated shifts (where black women had higher rates of screening) invasive squamous cell carcinoma incidence rates among black women were still higher than those of white women. These differences can be accounted for due to differences in the quality of the screening and even management of positive screens, as well as potential risk factors of squamous cell carcinoma by race [3].

When examining survival rates for cervical cancer, Abdallah et al. found that White patients diagnosed with a localized cervical cancer in Alabama had higher relative survival rates compared to Black patients [4]. Other studies also have shown that although the survival gaps between White patients and Black patients narrowed from 13% to 8% between 1983 to 2012, when hazard ratios were calculated, Black race, older age, and medium–high poverty groups were associated with lower survival over a span of 30 years [5]. It has also been reported that compared to White patients, one-year survival was 3.5% lower in Black women, five-year survival was 8% lower in Black women, and a higher proportion of Black patients had distant-stage cancers [6]. In patients with stage IVB cervical cancer, the African American race was associated with an overall poorer survival compared with the White race [7]. Additionally, it has been shown that Asian American women had a better overall survival than White women [8].

An important factor in survival for cervical SCC is tumor grade at time of diagnosis and the effect it can have on management of disease. According to Matsuo et al., tumor grade is a prognostic factor for SCC, and higher-grade tumors are associated with decreased survival [9]. Screening is an important part of detecting and diagnosing cervical SCC early. While the incidence of cervical SCC has been decreasing in all races and all age groups due to screening for cervical SCC with the Papanicolaou (Pap) smear [10], when adjusting for age, the incidence of cervical cancer increased with age for non-Hispanic Blacks but decreased for non-Hispanic Whites after the age of 50 [11]. Black women are more likely to be diagnosed at a later stage compared to White or Hispanic women [12]. Furthermore, Black women were older, had a higher rate of regional and distant metastasis, and received cancer-directed surgery less frequently [13]. According to Wang et al., despite screening associated shifts (where Black women had higher rates of screening) invasive SCC incidence rates among Black women were still higher than those of White women [3]. These differences may be accounted for due to differences in the quality of the screening, the management of positive screening tests, and risk factors of SCC by race [3].

While many studies have been conducted on the associations between race and tumor size at the time of cervical cancer diagnosis, and between race and cervical cancer survival, it is unknown whether race is an effect modifier between tumor size at the time of diagnosis and 10-year cervical SCC survival. 

Therefore, this study was conducted to study whether race plays a role in the tumor size at diagnosis and 10-year cancer-specific survival rates in women with cervical SCC. We hypothesized that race would modify the association between tumor size at diagnosis and 10-year cancer-specific survival rates.

## 2. Materials and Methods

### 2.1. Study Design

A non-concurrent cohort study that was a secondary analysis of data obtained from the Surveillance, Epidemiology, and End Results (SSER) Database for cervical SCC cases diagnosed from 2004–2018 was performed. SEER is a cancer surveillance program of the National Cancer Institute (NCI) that collects cancer incidence and survival data from population-based cancer registries that cover about 42.0% of the United States population, including regions from the Northeast, such as Connecticut, Massachusetts, New York, and New Jersey; regions from the South, such as Kentucky, Louisiana, Atlanta, Rural Georgia, Greater Georgia, and Texas; regions from the North Central, such as Iowa and Illinois; and regions of West, including Hawaii, New Mexico, Seattle–Puget Sound, Utah, San Francisco Oakland, San Jose–Monterey, Los Angeles, Greater California, Idaho, Arizona, Alaska, and Cherokee Nation [14].

### 2.2. Inclusion and Exclusion Criteria

Individual cases were included in this study if they were females diagnosed with cervical SCC with an ICD-0-3 Histology code between 8072–8078, in the United States between 2004–2018, had a known tumor size (millimeters (mm)), and had complete data for the variables in this study. Exclusion criteria comprised cases where individuals had two or more cancers, chronic end-stage diseases, recurrent cervical SCC, or missing information for variables in this study.

### 2.3. Study Population

There were 36,415 females, unique cases, diagnosed with cervical SCC in the SEER database. Amongst those patients, 7417 were diagnosed between 2004–2018 and were screened for study inclusion. This timeframe was utilized since 2004 was the first year that tumor size data were recorded in the SEER database. Amongst the 7417 participants diagnosed with cervical SCC between 2004–2018, 5106 had an unknown tumor size, 23 had inexact measures in mm, 138 had microscopic focus or foci only, and, in one case, the tumor/mass was not found, yielding 706 participants with valid information on the exposure of interest among the eligible participants.

### 2.4. Variables

The primary outcome variable for this study was the 10-year survival from cervical Squamous Cell Carcinoma. The main independent variable of this study was tumor size at diagnosis. The tumor size at diagnosis was divided into two separate categories based on mortality risk; 2–4 cm was high-risk, and <2 cm was low risk. Covariates obtained from the SEER database included age at diagnosis (years), race (White, Black, Asian/Pacific Islander, or Other), ethnicity (Hispanic or Non-Hispanic), marital status (single (never married), married, separated, divorced, widowed, unmarried, or domestic partner), income (less than $60,000 United States dollars or greater than or equal to $60,000 United States dollars), tumor grade (well differentiated, moderately differentiated, poorly differentiated, or unknown), tumor stage (localized, regional, or distant), and treatment (no surgery, local tumor excision or destruction, total hysterectomy (THR), total hysterectomy with bilateral salpingoopherectomy (THR w/BSO), radical hysterectomy, or hysterectomy not otherwise specified). Confounding variables such as smoking or other health conditions were controlled through randomization. 

### 2.5. Statistical Analysis

Stata version 15 (College Station, TX, USA) was utilized to conduct data analysis. A descriptive analysis was conducted to attain an understanding of the data, check for the distribution of variables, and determine if there was any missing information in the SEER database. Next, we conducted a bivariate analysis to identify any confounders, utilizing Chi-Square tests for categorical variables (*p* < 0.01) and T-Tests for continuous variables (*p* < 0.01). After that, a collinearity diagnostic analysis was performed to assess the degree of correlation between variables. Finally, unadjusted and adjusted Cox regression analyses were conducted to calculate hazard ratios and their corresponding 95% confidence intervals (CI).

### 2.6. Ethical Issues

Our study had a limited sample size due to a profound amount of missing information on tumor size in the SEER database, especially for minority races. Therefore, this study lacked adequate power to determine all potentially significant findings. Given the limited number of patients in the SEER database with tumor size data, especially patients belonging to a racial minority, it must be made a priority to have this information collected and accessible to have accurate information for further analysis and implementation of recommendations to improve health and survival.

## 3. Results

Table 1 shows the baseline characteristics of the study participants according to tumor size. Amongst women in our study with cervical SCC, who had tumors 2–4 cm large at the time of presentation, a statistically significantly higher proportion of those women were Black and Asian/Pacific Islander when compared to women presenting with smaller tumor sizes (*p*-value 0.007). Additionally, in patients with larger tumor sizes, a significantly higher proportion of patients did not undergo surgery (21.1%) compared with women with smaller tumor sizes (1.2%), whereas women with smaller tumor sizes had a statistically significantly higher percentage of local tumor excision/destruction (23.9%), THR (13.8%), and THR w/BSO (22.3%) compared with women with larger tumor sizes (10.7%, 2.9%, and 14.8%, respectively). There was a significantly higher proportion of women with poorly differentiated tumors compared with women with smaller tumor sizes (*p*-value < 0.001), whereas these women also had well differentiated tumors compared with women with larger tumor sizes. Furthermore, there was a lower proportion of women with a localized stage compared to women with smaller tumor sizes (*p*-value < 0.001). Women with larger tumor sizes had a higher proportion of patients with regional metastasis compared to women with smaller tumor sizes (42.8% vs. 8.7%; *p*-value: <0.001). Lastly, the year at diagnosis, ethnicity, and income did not show statistically significant differences in distribution according to tumor size.

Table 2 reveals the predictors of survival women with cervical SCC. The interaction between race and tumor size was not statistically significant, thus the data were not stratified according to race/ethnicity. Tumor size and stage of cancer were the only covariates that were statistically significantly associated with survival in our study population. The adjusted hazard of death in patients with larger tumor sizes (2–4 cm) was 2.7 times higher than those with a tumor size of <2 cm (HR: 2.7; 95% CI: 1.3–5.8). Stage of cancer revealed that regional cervical SCC (HR: 5.6; 95% CI: 2.7–11.4) had a 5.6 increased hazard of death, and distant metastasis (HR: 9.6; 95% CI: 3.3–28.29) was associated with a 9.6 increase in mortality as compared with localized cervical SCC. Age was not associated with mortality (HR: 1.0; 95% CI: 1.0–1.0). Similarly, no statistically significant associations for race, ethnicity, diagnosis year, treatment options, and income were observed with 10-year cancer-specific survival.

Figure 1 presents the Kaplan-Meier survival estimates. There was a statistically significant difference in survival between patients with tumor size < 2 cm compared with those with a tumor size between 2−4 cm (*p*-value < 0.001)

## 4. Discussion

While our data found that Black/Asian/Pacific Islander patients presented with a significantly higher proportion of larger tumors as compared to smaller tumors for cervical SCC, and that larger tumors were associated with decreased 10-year-cancer-specific survival, our data found that race did not modify the association between tumor size and 10-year cancer-specific survival in women with cervical SCC in the US. In other words, the survival did change at respective tumor sizes depending on the race of the patient. Such a finding could have been due to the large amount of missing data, especially for minority races, and having a small sample size overall. Additionally, a patient’s stage of cancer was statistically significantly associated with survival in our study population. Patients with regional or distant metastases also had a higher risk of death compared with patients with localized metastases.

While our study did not reveal that race was an effect modifier between tumor size and survival, it did, however, observe similar prognostic factors associated with poor survival in women with cervical SCC such as larger tumor size at diagnosis, older age at diagnosis, higher grade, etc. Currently, the literature reports several prognostic factors that influence survival rates in women with cervical SCC [15]. For example, Hu et al. revealed that higher stage, regional and distant metastases, larger tumor size, and higher grade all had significant adverse effects on women with cervical SCC [15], which paralleled the findings from our study as well. Additionally, racial disparities in cervical cancer incidence and mortality have been persistent for many years in the United States. For example, Abdalla et al. demonstrated that White women diagnosed with localized stages of cervical cancer in Alabama always had better chances of survival [3]. This was because their relative survival rates were always more than 77%, compared to Black women with cervical cancer [3]. This is different from our study because, after adjusting for confounders, race did not seem to influence the association between tumor size and survival. This could largely be because our sample size was significantly reduced after inclusion, and exclusion criteria were applied. It could also be due to the abundance of missing information for Black women as well as women of Asian/Pacific Island descent in the SEER database.

Cancer incidence and overall survival outcomes among different races and ethnic groups can be considerably different due to the lack of standardization in access and quality of health care, especially when considering early detection, prevention, and up to date treatment [16]. When looking at screening and treatment, Black women were more likely to be diagnosed at a later stage compared to White or Hispanic women [16]. Furthermore, Black women were older and had a higher rate of regional and distant metastasis, as well as received cancer-directed surgery less frequently [11,12]. Many studies have found that older age, Black race, and medium–high poverty groups were associated with less survival [3,4,5,6,17]. Overall, most of the studies that assessed the association between race and survival, or race and incidence rates of cervical squamous cell cancer, have shown that, although incidence rates have been decreasing equally among the different races, survival rates and overall survival are still less in Black patients as compared to White, Hispanic, and Asian patients. A recent study sound that increased incidence rates of cervical SCC in Black women were prevalent in all socioeconomic stratas across the board [17]. Many studies suggest a difference in disease etiology, such as differences in HPV genotypes and variants, or lack of effective screening, access to quality healthcare, and treatment to explain differences between races, especially underserved minorities [3,13]. For example, the HPV35 variant is not covered in the vaccinations currently available. However, the HPV35 variant is disproportionately more strongly linked to cervical carcinogenesis in African American women compared to women of other races and ethnic groups [18]. Regional and racial/ethnic disparities in health outcomes are often associated with socioeconomic factors such as socioeconomic status, neighborhood poverty, and other social determinants of health [10]. Lastly, in regard to survival rate and tumor grade, tumor grade is a prognostic factor for squamous cervical cancer, particularly early-stage disease [8]. In this population, higher tumor grade is associated with decreased survival and poorer prognosis. Therefore, routine evaluation of tumor grade with synoptic description is highly recommended in daily practice.

Multiple covariates, including tumor size, race, age, treatment modalities, stage of cancer, year at diagnosis, ethnicity, and income, were analyzed to determine associations between race, tumor size, and survival in women with cervical SCC. Only tumor size at diagnosis and stage of cancer were found to have a significant impact on overall survival. Unlike other studies that found a significant decrease in overall survival when tumor size was >4 cm in diameter, our study uniquely found an decreased risk of survival (increased risk on mortality) in tumors measuring just 2–4 cm as well [19]. Our study further validated prior studies that demonstrated lymph metastasis and anatomic extension of the metastatic site with increased the risk of death [20]. While the International Federation of Gynecology and Obstetrics staging system is used most often to stage gynecological cancers, including cervical cancers, the SEER database is unique in that it groups cancers into localized, regional, and distant stages. Further analysis on the interactions of stage and grade and their combined effects on cancer-specific survival as a prognostic factor for cervical SCC should be explored [8]. Nonetheless, these findings only further emphasize the importance of screening and early detection of cervical cancer to ensure a good prognosis and improve overall survival.

This study entails several potential limitations that warrant cautious interpretation of our findings. Most notably, our study grappled with a restricted sample size, primarily attributable to the substantial amount of missing data concerning tumor size within the SEER database, particularly among minority racial groups. Consequently, our study was constrained in its ability to achieve adequate statistical power, which limited our capacity to detect potentially significant findings. Additionally, the database failed to include information on radiation therapy (RT), which is considered the standard of treatment for cervical SCC and may alter survival outcomes [21]. Furthermore, it is imperative to acknowledge that the study’s design, specifically its non-concurrent cohort approach, imposes certain constraints on its capacity to establish causal relationships definitively. Furthermore, the extent to which the study’s findings can be generalized may be circumscribed, primarily stemming from the database’s inherent limitations. This dataset was derived exclusively from 13 states, potentially skewing the representation of diverse populations, especially in relation to non-White females, which might impact the external validity of our findings. Lastly, it is worth noting that the relatively brief timeframe for data collection, spanning from 2004 to 2018, may have inadvertently limited the inclusion of patients in the database, consequently influencing our ability to achieve the necessary statistical power for robust analysis and comprehensive insights.

## 5. Conclusions

This study’s findings elucidate a multifaceted scenario. Notably, it was observed that a higher proportion of Black, Asian, and Pacific Islander patients exhibited tumor sizes falling within the 2–4 cm range, as opposed to those with tumor sizes less than 2 cm. Intriguingly, however, despite this disparity in tumor size, individuals within the 2–4 cm category experienced a markedly decreased 10-year survival rate. Additionally, patients with regional or distant metastasis also faced diminished 10-year survival prospects. It is crucial to underscore that, in the context of these findings, race did not emerge as a modifier of the interaction between tumor size and 10-year survival rates. Nevertheless, the findings should be considered within the context of certain limitations inherent in the study, including missing data, brief timeframe on data collection, lack of RT information and, most notably, the limited sample size of patients in the SEER database, especially those from racial minority groups, remaining a significant caveat. Considering this, there arises an imperative need to prioritize the systematic collection and accessibility of tumor size data to ensure the availability of accurate and comprehensive information for future analyses and the formulation of recommendations aimed at enhancing health outcomes and survival rates. Future studies may assess the association between race and tumor size at diagnosis and 10-year survival using larger data sets such as the National Cancer Data Base to see whether an association exists. 

Furthermore, the study underscores the necessity for more extensive exploration of various risk factors that may exert an influence on survival rates. This includes a thorough investigation into potential disparities in screening practices, treatment modalities, management strategies, vaccination rates, as well as the plausible existence of biological and pathophysiological distinctions among different demographic groups. Such comprehensive explorations are pivotal in fostering a deeper understanding of the complex interplay of factors impacting health outcomes and survival.

## Figures and Tables

**Figure 1 ijerph-20-06742-f001:**
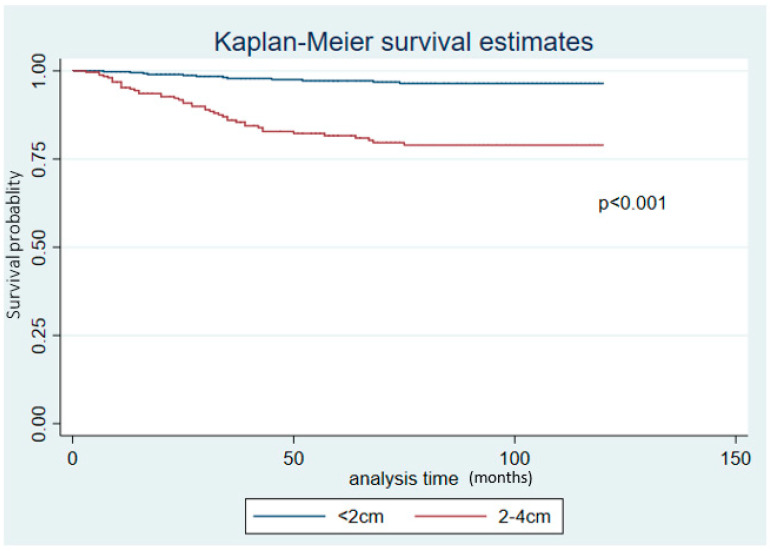
Kaplan–Meier survival estimates for tumor sizes < 2 cm and 2–4 cm.

**Table 1 ijerph-20-06742-t001:** Baseline Characteristics of the Study Participants According to Tumor Size.

	Independent Variable				
Characteristics	Tumor Size < 2 cm		Tumor Size between 2–4 cm		*p*-Value
	n	%	n	%	
**Age-mean (SD ^1^)**	435	43.5 (12.2)	271	48.5(14.1)	**<0.001**
**Race**					0.007
White	338	77.7	102	37.6	
Asian/Pacific Islander	49	11.26	48	17.71	
Black	48	11.03	41	15.13	
Missing	0	0	80	29.5	
**Ethnicity**					0.442
Non-Spanish Hispanic	366	84.14	222	81.92	
Spanish Hispanic–Latino	69	16.86	49	18.08	
Missing	0	0	0	0	
**Income**					0.507
<$60,000	109	25.06	74	27.31	
≥$60,000	326	74.94	196	72.32	
Missing	0	0	1	0.37	
**Treatment**					**<0.001**
No surgery	8	1.84	57	21.03	
Local tumor excision/destruction	104	23.91	29	10.70	
THR ^2^	60	13.79	8	2.95	
THR w/BSO ^3^	97	22.3	40	14.76	
Radical hysterectomy	143	32.87	128	47.23	
Hysterectomy NOS ^4^	23	5.29	8	2.95	
Missing	0	0	1	0.37	
**Grade**					**<0.001**
Well differentiated	44	10.11	7	2.58	
Moderately differentiated	133	30.57	78	28.78	
Poorly differentiated	110	25.29	134	49.45	
Unknown	142	32.64	50	18.45	
Missing	6	1.38	2	7.38	
**Stage**					**<0.001**
Localized	395	90.8	143	52.77	
Regional	38	8.74	116	42.8	
Distant	2	0.46	12	4.43	
Missing	0	0	0	0	
**Year at diagnosis**					0.394
2004–2011	247	56.78	145	53.51	
2012–2018	188	43.22	126	46.49	
Missing	0	0	0	0	

Legend: ^1^ Standard deviation; ^2^ Total Hysterectomy; ^3^ Bilateral Salpingo-Oophorectomy; and ^4^ Not Otherwise Specified. Bolded *p*-values indicate statistical significance.

**Table 2 ijerph-20-06742-t002:** Associations Between Patient Demographics and Cancer Characteristics with 10-Year Cancer-Specific Survival in Women with Cervical Squamous Cell Carcinoma.

Characteristics	Unadjusted	Adjusted
	HR ^1^ (95% CI ^2^)	HR (95% CI)
**Tumor Size**		
<2 cm	Reference	Reference
2–4 cm	6.8 (3.6–12.8)	2.68 (1.25–5.77)
**Race**		
White	Reference	
Black	1.10 (0.49–2.46)	
Asian/Pacific Islander	1.33 (0.65–2.76)	
**Age**	1.04 (1.02–1.06)	1.01 (0.99–1.03)
**Treatment**		
No Surgery	8.43 (3.34–21.27)	2.40 (0.93–6.20)
Surgery	Reference	Reference
THR ^3^	0.85 (0.21–3.41)	2.76 (0.63–12.11)
THR w/BSO ^4^	1.75 (0.63–4.80)	2.44 (0.86–6.91)
Radical hysterectomy	1.35 (0.53–3.42)	1.18 (0.46–3.04)
Hysterectomy NOS ^5^	0.75 (0.89–6.19)	1.05 (0.12–8.87)
**Stage**		
Localized	Reference	
Regional	9.98 (5.45–18.31)	5.57 (2.72–11.42)
Distant	18.52 (7.17–47.76)	9.60 (3.25–28-29)
**Year**		
2004–2011	Reference	
2012–2018	1.27 (0.73–2.21)	
**Ethnicity**		
Non-Spanish Hispanic-Latino	Reference	
Spanish Hispanic-Latino	1.19 (0.60–2.37)	
**Income**		
<$60,000	Reference	
≥$60,0000	0.68 (0.39–1.18)	

Legend: ^1^ Hazard Ratios; ^2^ Confidence Intervals; ^3^ Total Hysterectomy; ^4^ Bilateral Salpingo-Oophorectomy; and ^5^ Not Otherwise Specified.

## Data Availability

The data for this project was obtained from the Surveillance, Epidemiology, and End Results (SEER) Database, which is part of the National Cancer Institute (NCI) Division of Cancer Control and Population Services (DCCPS).

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
