# Peer review of "Effect Modification of Race on the Associated Tumor Size at Diagnosis and 10-Year Cancer Survival Rates in Women with Cervical Squamous Cell Carcinoma in the United States"

_ijerph, 2023, doi:10.3390/ijerph20186742_

Round 1

Reviewer 1 Report

The paper aims to evaluate whether or not race plays a significant part in overall survival of SSC patients. The idea is sound, but as the authors pointed out themselves, once the necessary adjustments in sample size were made, it just isn't large enough to see a significant difference, which is not the same as saying there isn't one.

The question that comes to mind is why not use the much larger NCDB? Was it a matter of data quality or access? With a larger database, it might have been possible to end up with a suitable sample size even after the necessary adjusments.

Reviewer 2 Report

Dear editor and dear authors,

The manuscript addresses an essential thematic area regarding its consequences on primary prevention and developing new politics.  

Some Keywords are not indexed in the Mesh. I suggest reviewing that.

 The introduction is well-framed. The authors explained the relevance of the present study and justified the theme. 

The study design, inclusion and exclusion criteria, study population, variables and statistical analysis are well-framed in the materials and methods section. Despite being at the end of the manuscript, I would add the ethical issues at the end of this section.

In the results, the authors present Table 1; when reading the table, we realized that some characteristics are missing, a suggest adding a row for missing attributes, for example in the race, the authors have n=271 for tumor size between 2-4 cm, white (n=102), Asian/Pacific Islander (n=48) and black (n=41) in total n=252 so we have 19 missing values. 

This study has a limitation related to the sample size, and the author identified that. In conclusion, the authors enhanced the relevance and implication of the study.

Reviewer 3 Report

The research paper is a study of retrospective cohort study done on female patient with cervical cancer. The study examined race of patients as a variable contributing to the size and the survival of the patients. The study is interesting and falls within the scope of the journal and recommend publication considering the following recommendation:

-        In the abstract the conclusion section does not reflect the result (please rephrase)

-        In the methodology:

o   The methodology should clearly have stated how the study dealt with confounder (example smoking) including other disease

o   P significant value should be stated in the statistical part of the methodology part

o   Put the title the x-axis and the y axis of the figure

o   In the discussion part please defend more elaborately why race was not significant while them mentioned results did show significant difference

o   Study limitation of the must be clearly stated in the conclusion part

o   Other minor comments:

§   Review again the whole manuscript for typing and grammatical mistakes

§  Please double check the reference style so it has all the assume style either the full name or the abbreviated name of the journal

Minor changes

Round 2

Reviewer 1 Report

The main problem with the paper (sample size) remains and sadly cannot be properly addressed at the moment, as the authors pointed out. Ideally, the study would have used the larger NCDB for data, but aside from that being a whole different study, the author don't hae access to it. Nevertheless, the authors properly aknowledge that flaw in the manuscript and correctly point to the necessity of better data curation on cancer databases.